# Impact on the Transcriptome of Proton Beam Irradiation Targeted at Healthy Cardiac Tissue of Mice

**DOI:** 10.3390/cancers16081471

**Published:** 2024-04-11

**Authors:** Claudia Sala, Martina Tarozzi, Giorgia Simonetti, Martina Pazzaglia, Francesco Paolo Cammarata, Giorgio Russo, Rosaria Acquaviva, Giuseppe Antonio Pablo Cirrone, Giada Petringa, Roberto Catalano, Valerio Cosimo Elia, Francesca Fede, Lorenzo Manti, Gastone Castellani, Daniel Remondini, Isabella Zironi

**Affiliations:** 1Department of Medical and Surgical Sciences (DIMEC), Alma Mater Studiorum University of Bologna, 40127 Bologna, Italy; claudia.sala3@unibo.it (C.S.); gastone.castellani@unibo.it (G.C.); 2Biosciences Laboratory, IRCCS Istituto Scientifico Romagnolo per lo Studio e la Cura dei Tumori (IRST) “Dino Amadori”, 47014 Meldola, Italy; giorgia.simonetti@irst.emr.it (G.S.);; 3Institute of Bioimaging and Molecular Physiology, National Council of Research (IBFM-CNR), 90015 Cefalù, Italygiorgiorusso@lns.infn.it (G.R.); 4Laboratori Nazionali del SUD, National Institute of Nuclear Physics, (LNS-INFN), 95125 Catania, Italygiada.petringa@lns.infn.it (G.P.);; 5Department of Drug Science, Section of Biochemistry, University of Catania, 95125 Catania, Italy; racquavi@unict.it; 6Department of Physics “E. Pancini”, University of Naples Federico II, 80126 Naples, Italy; valeriocosimo.elia@unina.it (V.C.E.); fede@na.infn.it (F.F.); manti@na.infn.it (L.M.); 7National Institute of Nuclear Physics, Napoli Section (INFN NA), 80126 Naples, Italy; 8National Institute for Nuclear Physics, Bologna Section (INFN BO), 40127 Bologna, Italy; 9Department of Physics and Astronomy “Augusto Righi” (DIFA), Alma Mater Studiorum University of Bologna, 40127 Bologna, Italy

**Keywords:** proton beam therapy, radiotherapy, radiation effects, transcriptomics, cardiac tissue

## Abstract

**Simple Summary:**

The nature of different types of ionizing radiation is central to the modality of affecting biological targets. The main data library on radiotherapy effects we can access is on photon sources, and any other type of radiation is compared to that, not always considering that different physical features might contribute in quite different ways to the quality of visible effects. A large body of study already supports this vision, but a lot of work is still to be done, particularly on irradiated healthy tissue in the vicinity of the cancer target. This study aims to gain information on the effects of anti-cancer therapeutic protons as a function of radiation dose and time post-irradiation on healthy cardiac tissue through the analysis of transcriptionally activated genes and relative molecular pathways.

**Abstract:**

Proton beam therapy is considered a step forward with respect to electromagnetic radiation, thanks to the reduction in the dose delivered. Among unwanted effects to healthy tissue, cardiovascular complications are a known long-term radiotherapy complication. The transcriptional response of cardiac tissue from xenografted BALB/c nude mice obtained at 3 and 10 days after proton irradiation covering both the tumor region and the underlying healthy tissue was analyzed as a function of dose and time. Three doses were used: 2 Gy, 6 Gy, and 9 Gy. The intermediate dose had caused the greatest impact at 3 days after irradiation: at 2 Gy, 219 genes were differently expressed, many of them represented by zinc finger proteins; at 6 Gy, there were 1109, with a predominance of genes involved in energy metabolism and responses to stimuli; and at 9 Gy, there were 105, mainly represented by zinc finger proteins and molecules involved in the regulation of cardiac function. After 10 days, no significant effects were detected, suggesting that cellular repair mechanisms had defused the potential alterations in gene expression. The nonlinear dose–response curve indicates a need to update the models built on photons to improve accuracy in health risk prediction. Our data also suggest a possible role for zinc finger protein genes as markers of proton therapy efficacy.

## 1. Introduction

Proton beam therapy (PBT) is today considered an advanced radiotherapy (RT) procedure based on high-energy photons/electrons. This is due to the physical properties of accelerated charge particles, whose inverse dose–depth profile (Bragg curve) considerably spares the organs at risk (OARs) by delivering a lower dose [1]. This is considered the most promising for cardioprotection from radiation-induced toxicity in breast cancer (BC) treatments [2]. Indeed, based on long-term follow-up data, it has been shown that PBT can improve both progression-free survival and reduce breast cancer mortality [3,4,5]. The potential impairment of radiation-induced side effects, such as risks of cardiac toxicity [6], is due to the fact that PBT delivers the lowest mean heart dose (MHD) of any conventional photon technique [7,8].

Although many of the response mechanisms to ionizing radiation (IR) at the cellular level are mainly driven by the modality of energy deposition at the nanometer scale (e.g., LET or linear energy transfer), some unique effects have been reported for protons [9]. A recent study compared the genomic response of the mouse aorta to proton and gamma whole-body radiation following increasing doses from 0.5 to 200 cGy [10], detecting marked differences in the genomic response. Another investigation showed that for high-charge-and-energy (HZE) particles or gamma irradiation (γ-IR), there is not a clear lower IR threshold and that they share 12 twofold differentially expressed genes (DEGs). These 12 genes predicting various degrees of cardiovascular, pulmonary, and metabolic diseases, cancer, and aging revealed a nonlinear DEG pattern in particle IR-exposed hearts, whereas the majority of γ-IR-exposed hearts revealed a linear pattern of DEGs [11]. Interestingly, both protons and electron beams follow the dose–response curves for the induction of DNA double-strand breaks (DSBs) with a linear dose-related increment, also observed for photon radiation [12], expressing only more highly localized and clustered DNA damage from particle radiation compared to X- and γ-rays [13]. Therefore, the now-established models of cardiovascular risk based on photon radiation may not accurately predict the risk associated with PBT.

This study aims to expand current knowledge on possible proton-associated cardiovascular risk along the dose–response curve in the range used for oncological PBT on healthy heart tissue with an “omics” approach. The hearts of orthotopic xenograft murine models, subcutaneously inoculated with human breast cancer cells, were collected after 3 and 10 days following proton irradiation delivered as in a clinical scenario. The protocol design and the analysis have been projected to follow the gene expression at an early and a later stage (10 days after exposure), with a specific focus on those responsible for cardiotoxicity. Gene expression analysis by microarray was performed to study transcriptionally activated genes, molecular pathways, and cellular networks.

## 2. Materials and Methods

### 2.1. Ethics Statement and Animal Model

The experiments were performed in accordance with a European Council directive and Italian regulations (2010/63/EU and D.Lgs. 26/2014). The project was approved by the Italian Ministry of Health (authorization 527/2016-PR, approved on 26 May 2016). Efforts were employed to replace, reduce, and refine the use of laboratory animals. To avoid unnecessary suffering of treated mice, euthanasia was performed as soon as the final score was reached. The endpoint used to determine if animals should undergo euthanasia was reached when tumor lesions showed a dimension higher than 1.2 cm and/or weight loss more than 20%. All reasonable efforts were made to ameliorate suffering, avoiding the most painful procedures. To minimize suffering and mouse distress, standard environmental enrichment of two nestles, a cardboard Fun Tunnel, and one wooden chew block was provided.

Experiments were performed as shown in Figure 1 on 8-week-old BALB/c nude female mice (Charles River Laboratory) weighing 24 ± 3 g. Animals were housed in IVC cages at constant temperature (23–25 °C) under a 12/12 h light/dark cycle with ad libitum access to food and water. Mice were housed using a stocking density of three mice per cage in individual IVC cages. A total of 4 × 10^6^ MDA-MB-231 BC cells were inoculated in a group of 24 BALB/c nude mice into the mammary fat pad [9,14,15]. Animal health and behavior were monitored twice a week together with body weight and clinical specific signs up to euthanasia.

### 2.2. Animal Radiation Treatment

After two weeks of growth, the tumors had reached a size of 8 ± 2 mm, monitored by a digital caliper. Inoculated mice were divided randomly into four groups of six: three groups for proton irradiation at 2, 6 and 9 Gy (D2, D6 and D9, respectively) and one for a non-irradiated control group (CTRL). Proton irradiation was performed in two different daily sections at the PBT CATANA (Centro di AdroTerapia e Applicazioni Nucleari Avanzate) facility of the Istituto Nazionale di Fisica Nucleare INFN-LNS, Catania, Italy. Correct positioning of the animals to localize the region tumor in the center of the SOBP (spread-out Bragg peak) was carried out using a positioning system formed by a light field and a laser for the identification of the isocenter, and was verified by radiographic images and small metal clips integral with the tumor region. The beam energy (62 MeV) was set to irradiate from the skin to the heart included. The collimator and thus the transverse shape of the beam was circular with a diameter of 15 mm. The spatial extension of the proton SOBP therefore covered the entire tumor region and the underlying healthy tissue. The estimated dose reaching the hearts was about 2, 6 and 9 Gy ± 3%. The prescribed dose was released in a single session, with a dose rate between 0.7 and 2 Gy/min. For each group, three randomly chosen mice were euthanized at 72 h (early stage, T3) or 10 days (late stage, T10) post-PT treatments. Whole hearts were collected and stored at −80 °C until molecular analyses [16].

### 2.3. RNA Extraction and Microarray

Frozen whole cardiac tissue samples were homogenized in TRIzol (Invitrogen, Carlsbad, CA, USA) and RNA was extracted according to the manufacturer’s recommendations. Preparation of labeled single-stranded complementary DNA (ss-cDNA) was performed from 100 ng RNA, as described previously [17]. Three independent samples of each condition (except for the 10-day control—two samples only) were hybridized to mouse Clariom D arrays (Thermo Fisher Scientific, Waltham, MA, USA) according to the manufacturer’s recommendations. This technology allows the detection of transcriptome-wide variations of gene expression at exon resolution, thus also allowing resolution of rare transcripts and alternative splicing events while providing insights on long-noncoding RNAs, as well as increasing the probability of identifying complex disease signatures.

### 2.4. Computational Analysis

#### 2.4.1. Data Preprocessing

Raw CEL files were processed using the R library oligo [18]. The extracted intensity values were normalized using the robust multichip average (RMA) algorithm [18] with the option “target = core” to use transcript clusters containing “safely” annotated genes [19]. Annotations were retrieved using the R library mta10transcriptcluster.db. Transcript clusters that mapped multiple gene symbols and control probes were removed, and the expression values were log-transformed for the statistical analysis, as detailed below.

#### 2.4.2. Differential Expression Analysis

Principal component analysis (PCA) was performed using the prcomp function of the R library factoextra, considering the RMA-normalized data. Differential expression analysis between each pair of treatment doses (2 Gy vs. CTRL, 6 Gy vs. CTRL, 9 Gy vs. CTRL, 6 Gy vs. 2 Gy, 9 Gy vs. 2 Gy, and 9 Gy vs. 6 Gy) was performed using the R library limma function [20]. Specifically, after computing a linear regression model for each gene (using the lmFit function), moderated t-statistics were computed by empirical Bayes moderation of the standard errors towards a common value (using the eBayes function). *p*-values were adjusted for multiple testing using the Benjamini–Hochberg procedure. For each comparison, a volcano plot representing the test’s statistical significance (−log10 (*p*-value)) versus the magnitude of the log2 fold change (LFC) was produced using the function volcano plot of the R library limma.

#### 2.4.3. Functional Enrichment Analysis

To gain more functional insight into the differentially expressed genes, functional enrichment analysis was performed with an overrepresentation analysis approach. DEGs at the different dosages were considered separately, and for each class, over-expressed genes (LFC > 0) and under-expressed genes (LFC < 0) were considered separately. Functional analysis was performed using the Bioconductor (https://bioconductor.org/ (accessed on 22 February 2024)) package “gprofiler2” [21] on Gene Ontology categories and on the KEGG database.

## 3. Results

### 3.1. Differential Expression Analysis

According to the box plots obtained on the preprocessing analysis of the 23 analyzed samples (Appendix A), the dataset was substantially homogeneous in terms of expression value distribution for each sample. PCA (Appendix A) showed a partial grouping of the samples by dose based on their global expression profile, and only a slight separation of samples by time. Overall, samples irradiated at 2 and 6 Gy are closer in the PCA plots compared to 9 Gy-exposed samples, suggesting a higher similarity. DEGs relative to each dose and their overlapping between the three doses are reported in volcano plots comparing irradiated and control samples (Appendix A). Several DEGs were identified at T3, with a majority of them being over-expressed in the irradiated samples. Comparing 2 Gy vs. CTRL samples, we identified 219 DEGs, 205 of which were over-expressed in the irradiated samples. In 6 Gy vs. CTRL, we identified 1109 DEGs, 828 of which were over-expressed in the irradiated samples. For 9 Gy vs. CTRL, we identified 105 DEGs, 80 of which were over-expressed in the irradiated samples. However, at time T10, only one DEG (TC0M00000019.mm.1) was identified (in the 2 Gy vs. CTRL comparison), and it was under-expressed in the irradiated samples. Although this is a surprising result, we are confident in excluding potential experimental biases by having performed the treatment of the samples randomly and without procedural variations. Furthermore, the box plot in Appendix A confirms that the results obtained from each sample do not reveal the presence of outliers or macroscopic differences. Concerning the comparisons of samples irradiated with different doses, at T3, we identified 5371 DEGs when comparing the 9 Gy and the 6 Gy samples, while no difference was observed for the other doses. At T10, only a few genes were found to be differentially expressed when comparing 9 Gy and 2 Gy (11 transcripts) and when comparing 6 Gy and 2 Gy (1 transcript), while DEGs were identified when comparing 9 Gy and 6 Gy.

Overall, the comparison between each dose and the control samples identified 1183 genes that were differentially expressed (adj. *p*-value < 0.05) in at least one comparison at T3 (Figure 2A), 325 of which had a known gene name (according to the R mta10transcriptcluster.db database). On the other hand, only one transcript (with unknown gene name) was found to be differentially expressed between irradiated and control samples at T10 (Figure 2B).

To analyze the intensity of single gene expression as a function of proton doses at T3, we focused only on genes with an annotated name and with an adj. *p*-value < 0.05 compared to the controls. In addition, we selected genes with an expression rate or log fold change (LFC) above 1 or below −1. In Figure 3, the data collected from the 2 Gy dose is displayed in histogram form, accompanied by the log fold change (LFC) values in comparison to other doses. It is evident that a minimal number of genes (16) are significantly impacted (adjusted *p*-value < 0.05) by the lowest proton dose tested. Nonetheless, the graph also highlights that the predominant change was an increase in gene expression. The only genes showing an LFC < −1 were the protein-coding *Olfr192* and the small nucleolar *Snord85*. The figure does not include 13 genes that were over-expressed, as they are associated with predicted genes of unknown function, making their interpretation challenging. Notably, the data reveal an upregulation of a number of zinc finger proteins (ZFPs), with 8 out of 16 being ZFPs, which are known to be the most extensive group of transcription factors, thus having a significant impact on gene expression regulation. Additionally, the analysis indicates that doses of 6 and 9 Gy led to a similar level of over-expression for this gene group, as shown in Figure 3. However, for other genes, a log fold change (LFC) greater than 1 seen at 2 Gy was generally not replicated at the higher doses, pointing to a distinct dose-dependent response.

The dataset for the 6 Gy dose is represented by histograms in Figure 4, alongside the log fold change (LFC) values when compared to the other doses. The most striking observation is the substantial rise in the number of genes with an LFC greater than 1, totaling 60. However, among these, zinc finger proteins (ZFPs) accounted for only 9 out of 60, which is a smaller proportion relative to the data for the 2 Gy dose, and there was less overlap with the effects at the other doses. Only three genes exhibited an LFC less than −1 (*Mir6382*, *Mir883b*, and *Acot10*), and eighteen over-expressed genes were linked to predicted genes of unknown function, which were not included in the figure. Importantly, the majority of the protein-coding genes did not show the same increase level at the other two doses, indicating a dose-dependent effect.

The histogram for to the 9 Gy dose obtained from a LFC greater than 1 is shown in Figure 5. Unexpectedly, the number of over-expressed genes (nine) drops back to approximately the value observed at the lowest dose of 2 Gy (16). Proportionally, ZFPs are highly represented (four), and all of them are significantly over-expressed, even at 2 and 6 Gy. Six over-expressed genes refer to predicted genes of unknown function and are therefore excluded from the figure. Three genes showed an LFC < −1: the *Hspg2* gene coding for basement membrane-specific heparan sulfate proteoglycan core protein, which plays an essential role in angiogenesis and vascularization, the *Flnc* gene coding for filamin C, which plays a central role in sarcomere assembly and organization, and the *Ltbp4* gene coding for latent-transforming growth factor beta-binding protein 4, a TGFB binding protein essential for its role in the extracellular matrix and in maintaining elastic fiber properties in several tissue types, including muscular tissue [22].

These and some of the other protein-coding genes involved in pathways, such as cell cycle regulation, transcription regulation, cellular metabolism and vesicle trafficking, will be discussed in Section 3.2.

All DEGs (distinguishing LFC > 0 and <0) at the different proton doses were used as input for the functional enrichment analysis of the affected pathways (Table 1, Table 2a,b and Table 3). We observed that at all doses, the over-expressed genes caused a strong alteration of pathways involved in transcriptional regulation, specifically altering the DNA-binding activity of RNA polymerase II and transcription factors. This alteration is associated with the lowest *p*-value at 6 Gy and 9 Gy (adj. *p*-value = 0.0006 and 0.0009, respectively). At 6 Gy, a significant alteration in pathways involving energy metabolism was observed, which was mediated both by over-expressed genes (GO:0006119) and under-expressed genes (KEGG:05208). In this analysis, 6 Gy was the only dose that provided significant results associated with the under-expressed genes: relevantly, here we found a significant enrichment of pathways associated with cellular response to radiation.

### 3.2. Focus on Protein-Coding Genes

We focused on a subset of protein-coding genes listed in Appendix A, which play a role in key pathways pertinent to this study’s focus, namely, cell cycle or transcription regulation, cellular metabolism or vesicle trafficking, and cardiac tissue function. We tracked the expression levels of these genes across the different doses. Notably, these genes exhibited marked over- or under-expression when subjected to doses of 2, 6, and 9 Gy at three days post-irradiation. We categorized these genes based on their functional relationships to assess the influence of both dose and time elapsed since exposure on their expression patterns.

Figure 6 presents the log fold change (LFC) and the adjusted *p*-value from expression analyses at T3 (panels A and C) and T10 (panels B and D) at the three irradiation doses, specifically for the five genes associated with cell cycle or transcription regulation: Cdkn1a, Trp53inp1, Hsp90aa1, Eda2r, and Bhlhe40 (Appendix A). This representation highlights the bell-shaped trend of RNA expression for these genes, particularly evident at T3. As earlier described in this section, differences in gene expression at T10, although visible, did not show statistical significance.

Next, we focused on DEGs involved in cellular metabolism or vesicle trafficking (Figure 7), focusing on the genes Uprt, Lamp2, Ogn and Vamp7. Lastly, in Figure 8, we report the trend of gene expression alteration of Hspg2, Flnc and Ltbp4, genes that play relevant roles specifically in muscle cells or in the cardiac tissue. These genes also belong to the pathway KEGG:05205 (Table 3), which includes proteoglycans (PGs), key macromolecules in affecting tumor progression.

The predominant trend we observed was a nonlinear effect of the different PBT doses on gene expression alteration. Most genes showed an increased LFC at 6 Gy compared to the 3 Gy dose, which often represents the maximum value observed. Interestingly, the LFC value tends to decrease at 9 Gy compared to 6 Gy. With the only exception of uprt, at T10, we observe generally a more linear trend in gene expression levels, coherent with the substantial absence of statistically significant DEGs at this time point.

In Figure 9, the LFC of the same protein-coding genes discussed above is plotted as a function of time rather than doses, revealing that the change in the majority of over-expressed genes exposed to 2 Gy and 9 Gy is less at T3 than at 6 Gy, as we already know, but also that some of them are persistent over time, even if not in a significant way. In particular Cdkn1a, Uprt and Eda2r genes, coding for proteins involved in cell division cycle, in nucleotide metabolism and for a tumor necrosis factor receptor, respectively, appear to maintain the same degree of activation observed after 3 days also at a later time (after 10 days) only for the less effective doses (2 and 9 Gy). Interestingly, these results also showed that the LFC of osteoglycin (Ogn) follows a bell-shaped curve along the dose axis, while the effect of all three doses along the time axis is not persistent, no longer being over-expressed after 10 days from exposure. This non-structural matricellular protein is known to modulate cardiac inflammation, injury and function during viral myocarditis [23]. The level of Ogn expression has been correlated with heart hypertrophy, but has also been indicated to prevent the development of age-related diastolic dysfunction by reducing cardiac fibrosis and inflammation [24].

## 4. Discussion

In this work, we studied the transcriptional response of healthy cardiac tissue obtained from an orthotopic xenograft mouse model of breast cancer at an early and a later stage (10 days post-exposure) following proton beam irradiation with a spatial extension of the SOBP covering both the tumor region and the underlying healthy tissue at three different therapeutic doses. The Venn diagrams give clear evidence that at the early stage (3 days post-irradiation), several known and unknown genes were differentially expressed (adj. *p*-value < 0.05) compared to the control (CTRL). Notably, the data revealed a nonlinear, dose–response pattern of gene expression alterations resembling a bell-shaped curve. Further analysis of a select group of genes (protein-coding, with a single-fold increase or decrease in expression) at this early stage reinforced the bell-curve observation, indicating that the 6 Gy dose caused the most pronounced transcriptional changes. Among the differentially expressed genes (DEGs), zinc finger protein (ZFP)-coding genes were the most prevalent. ZFPs constitute the most extensive family of transcriptional regulators in mammals, with roles in DNA binding, RNA packaging, protein structure formation, lipid interactions, transcriptional activation, and apoptosis control [25,26,27]. Recent studies have proposed that some ZFP genes might act as oncogenes, contributing to the development and progression of cancer. However, ZFPs can also function as tumor suppressors. The role of ZFPs in cancer is complex and can vary depending on the specific ZFP and the context of its expression [28]. Other studies report some ZFP genes as markers of radioresistance, indicating a tumor’s ability to withstand radiation therapy. In this context, alterations in ZFP gene expression could contribute to the survival and repair mechanisms of cancer cells exposed to radiation, which might influence treatment outcomes [29].

Consequently, the functional enrichment analysis of DEGs performed in this study at the three therapeutic proton doses indicated a strong alteration in pathways involved in transcription regulation. The 6 Gy dose had a more prominent influence on the number of altered pathways than the 2 and 9 Gy doses. This intermediate dose significantly affected a broader spectrum of biological functions, involving also the cellular amide metabolic process, the establishment of protein localization, oxidative phosphorylation, drug metabolism via cytochrome P450, and myelin sheath chemical carcinogenesis with reactive oxygen species production. The 6 Gy dose also provided significant results in terms of functional enrichment from the few under-expressed genes that were associated with the cellular response to radiation pathways. Significantly, at the highest dose tested, the proteoglycan cancer pathway (KEGG:05205) stands out among the few pathways that are notably altered, primarily affecting transcription regulation. This pathway includes two significantly under-expressed genes (*Hspg2* and *Flnc*) and one that is over-expressed (*Cdkn1a*). In the context of cardiac tissue, the reduced expression of filamin C (*Flnc*) could be linked to impairments in cardiomyocyte contraction capabilities. The downregulation of heparan sulfate proteoglycan 2 (*Hspg2*) may disrupt normal cellular interactions and signaling pathways. Moreover, the upregulation of the cyclin-dependent kinase inhibitor 1A (*Cdkn1a*) might adversely affect the usual DNA replication and damage repair processes, potentially increasing the likelihood of muscle contraction failures.

In the literature, it is reported that the activation of *Cdkn1a* is followed by a series of events leading to G1-phase arrest through the inhibition of cyclin-dependent kinases 2, 4, and 6, which phosphorylate the RB protein [30]. An in vitro study conducted on human fibroblasts by Antoccia et al. [31] reported that exposure to protons led to an increase in the expression of this protein, although lower doses (1 and 2 Gy) and higher LET values were used. Noteworthily, Ricciotti et al. [10] found that for both genes *Cdkn1a* and *Eda2r*, there is an over-expression increasing with the dose until 2 Gy. In our work, we explore higher doses starting from 2 Gy, and this increasing trend for the expression of both is confirmed until 6 Gy. A further recent report [32] examining the transcriptome of mouse skin post-proton irradiation at 6 and 24 h using doses of 1 Gy and 0.1 Gy with a beam energy of 62 MeV/A found minimal modulation in genes controlled by DNA-damage checkpoints (such as *Cdkn1a*). Variation in gene expression has been sparsely documented in response to proton exposure, but there is currently a lack of in-depth studies specifically investigating the relationship between the variation in ZFP gene expression and the development of cancer in healthy tissue. Some studies focused on other proteins and genes. For example, a study conducted by Sertorio et al. [33] observed a similar change in the expression of the H*sp90aa1* protein in response to proton and XR irradiation, suggesting a possible role for this protein in the cellular response to radiation. Similarly, Nielsen et al. [34] reported similar values of *Tp53inp1* protein expression after proton irradiation, but in fibroblasts.

Subsequent analysis on a select array of protein-coding DEGs was conducted at both initial and later stages. As anticipated, an early-stage nonlinear response to varying dosages was observed, with the most significant changes occurring at 6 Gy. At the later stage, there were no statistically significant changes in gene expression in irradiated mice compared to untreated controls. At this time-point, genes associated with cell cycle and transcription regulation, as well as those involved in cellular metabolism and vesicle trafficking, exhibited a more consistent pattern of expression at T10 compared to T3. These findings collectively highlight a nonlinear response to PBT doses on gene expression. The data suggest that the response to increasing doses starts with a disruption of gene expression, particularly affecting the DNA binding of RNA polymerase II and transcription factors. This is followed by a shift at the intermediate dose that alters metabolic energy processes. At the highest dose, there is not only a continued change in transcription regulator activity but also an impact on genes directly involved in muscle cell function. It is quite surprising that significant differences in gene expression levels were not observable after 10 days post-irradiation. This might be due to turnover and repair mechanisms of cells exposed to radiation or to experimental reasons, such as a limited sample size that did not allow detection of smaller fluctuations in gene expression. Further studies are needed to provide a deeper understanding about repair mechanisms and radiotherapy side effects. This deeper understanding has significant implications for treatment optimization and patient care [35].

It is widely recognized within the scientific community that the reaction to radiation often exhibits a nonlinear dose–response relationship. Nevertheless, as thoroughly examined in a recent UNSCEAR report, even with an abundance of data covering diverse irradiation conditions and radiobiological outcomes, there remains a significant absence of agreement on definitive conclusions, particularly regarding transcriptomic changes. This is due to the intricate interplay of radiation-induced effects across a spectrum of low to moderate-high doses, varying dose rates, and the quality of the radiation [36].

While PBT is marginally more effective than photon therapy, with a constant relative biological effectiveness (RBE) of 1.1 for both cancerous and healthy tissue, the unique physical properties of protons render PBT dosimetrically superior for numerous treatment sites [35]. Recent studies have shown that RBE is not only spatially variable based on biological and physical factors [13,37,38] but may also exhibit varying thresholds according to the type of ionizing radiation (IR) and on a dose-dependent basis [11]. Specifically, the distinct physical characteristics of the proton beam delivery system, such as beam intensity, linear energy transfer (LET) and the spectrum of secondary particles, [39,40] are critical in creating differences in DNA-damage and -repair mechanisms when comparing PBT with traditional photon radiotherapy [41,42]. These results highlight the importance of further exploring how radiation influences gene expression, especially of ZFPs, and how this may contribute to the development or prevention of cancer in healthy tissue. Overall, this study sheds light on the effects of proton beam radiation on gene expression in cardiac tissue, offering valuable insights that can influence clinical practice.

A known limitation of this work is the lack of validation of the gene expression results with an independent methodology. These results, therefore, should be considered a starting point for further studies. In addition to the application of alternative methodologies to investigate transcriptional alterations in healthy tissue proximal to the target organ of the radiotherapy treatment, other approaches that could improve our collective understanding of the toxicity associated with such therapies might involve epigenomic regulation. For example, different dosages might differently affect the methylation profiles, which could be studied at a genome-wide level with the Infinium Mouse Methylation BeadChip. These data might be integrated with the dataset presented in this work to provide a more comprehensive understanding of the underlying causes of the transcriptomic response described in the present work. In addition to the bulk approach, other important insights might also come from single-cell studies, which could provide insight into cell-type-specific transcriptional response to radiotherapy treatment.

## 5. Conclusions

The data reported in this work show the transcriptional impairment of healthy cardiac tissue following proton beam irradiation targeted at breast cancer. Our results support a possible role of the ZFP genes as markers of radiotherapy side effects. In this work, we observed an unexpected nonlinear dose–response curve in several effector genes and transcriptional regulators, indicating the need for more in-depth experimental investigations on PBT aimed at updating the models built on photon performance that are not accurate enough to predict the risk associated with proton radiation. Furthermore, the total disappearance of the DEGs in the advanced phase post-treatment with protons at all the tested doses is a very interesting finding, suggesting that the cells of cardiac tissue have a notable ability to absorb an ionizing stimulus and avoid long-term changes in gene expression.

## Figures and Tables

**Figure 1 cancers-16-01471-f001:**
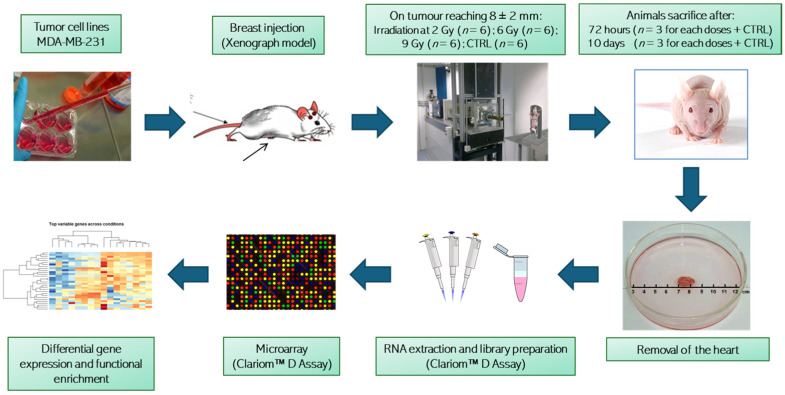
Graphical representation of the experimental workflow and analysis. Animal groups irradiated with different dosages were randomized and treated in two experimental rounds. Clariom™ D Assay from Thermo Fisher Scientific Inc., Waltham, MA, USA.

**Figure 2 cancers-16-01471-f002:**
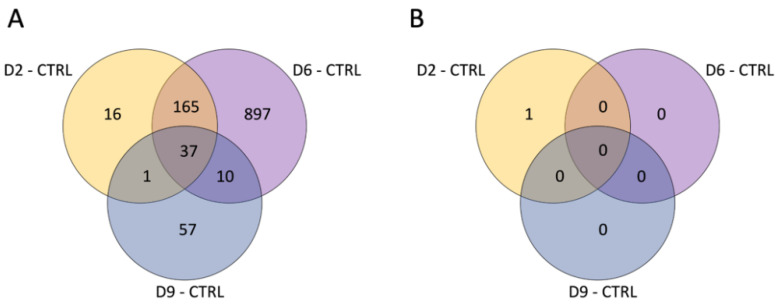
Venn diagram reporting the number of DEGs obtained in each comparison at T3 (**A**) and T10 (**B**).

**Figure 3 cancers-16-01471-f003:**
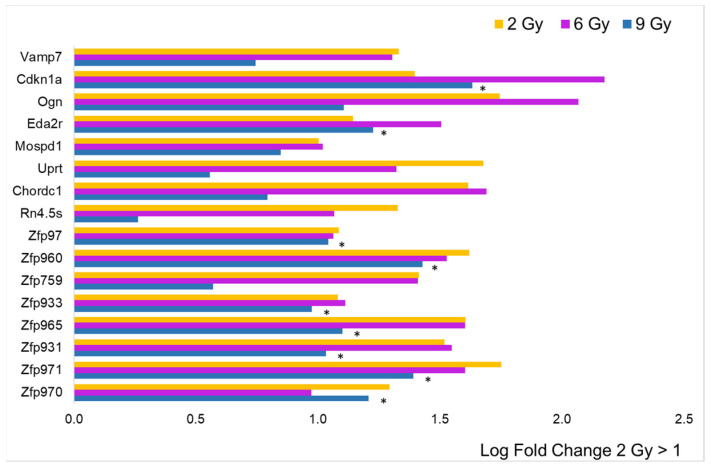
Comparison of radiation effects on gene expression obtained from the 2 Gy dose with those at the other two doses. The genes represented by the yellow bars (2 Gy) were selected based on LFC value (greater than 1, *n* = 16) from the sample expressing a significant difference compared to the CTRL. These same genes were all significantly different (adj. *p*-value < 0.05) compared to CTRL, even at 6 Gy (pink bars), while for those irradiated at 9 Gy (blue bars), the black asterisk indicates significance.

**Figure 4 cancers-16-01471-f004:**
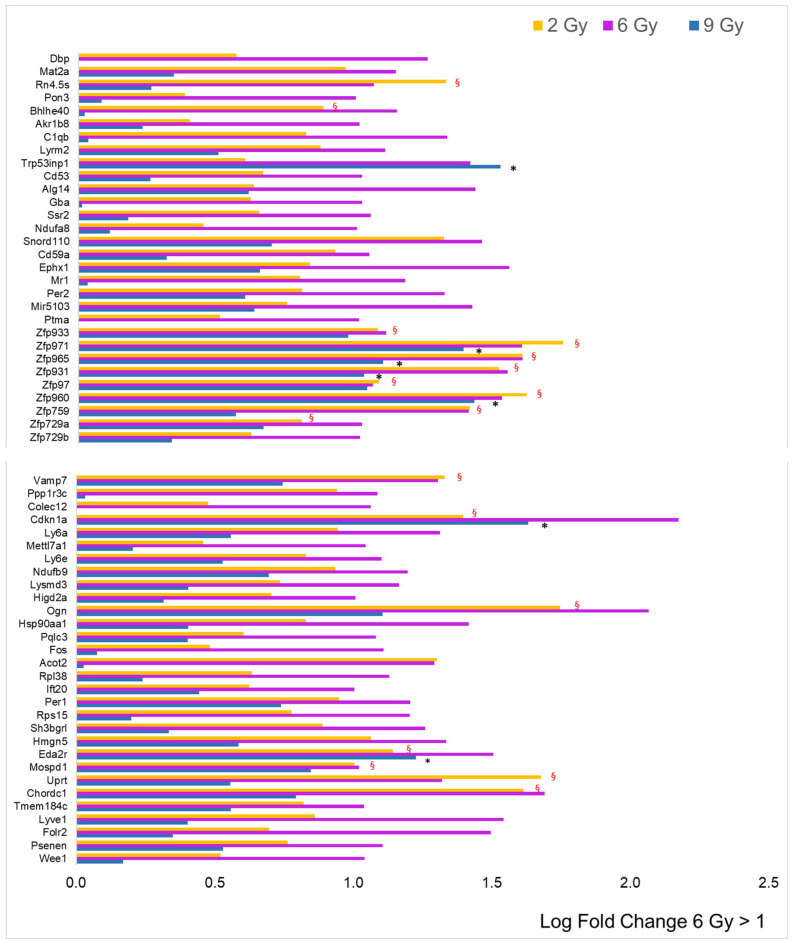
Comparison of radiation effects on gene expression obtained from the 6 Gy dose with those at the other two doses. The genes represented by the pink bars (6 Gy) were selected based on LFC value (greater than 1, *n* = 60) from the sample expressing a significant difference compared to the CTRL. Some of these genes were significantly different (adj. *p*-value < 0.05) compared to CTRL, even at 2 Gy (yellow bars, red § symbol) and/or at 9 Gy (blue bars, black asterisk).

**Figure 5 cancers-16-01471-f005:**
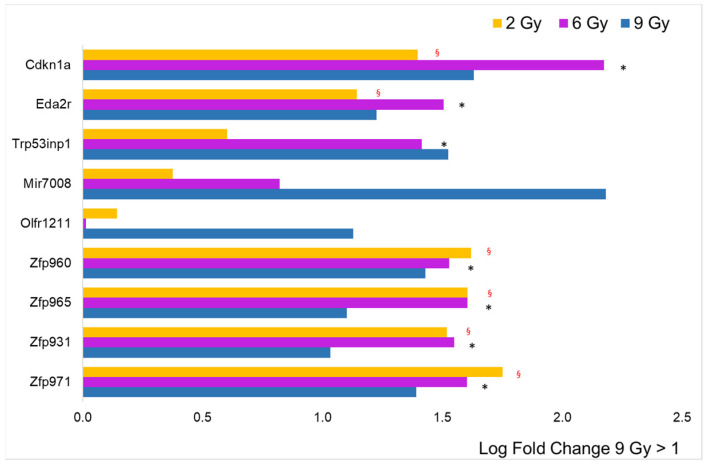
Comparison of radiation effects on gene expression obtained from the 9 Gy dose with those at the other two doses. The genes represented by the blue bars (9 Gy) were selected based on LFC value (greater than 1, *n* = 9) from the sample expressing a significant difference compared to the CTRL. Some of these genes were significantly different (adj. *p*-value < 0.05) compared to CTRL, even at 2 Gy (yellow bars, red § symbol) and/or at 6 Gy (blue bars, black asterisk).

**Figure 6 cancers-16-01471-f006:**
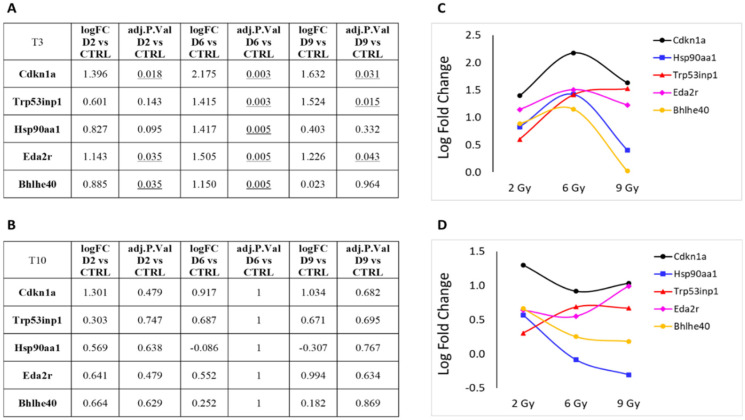
Protein-coding genes involved in cell cycle or transcription regulation pathways that showed significant over- or under-expression following exposure to 2, 6 or 9 Gy after 3 days (T3) from irradiation (**A**). (**B**) Same genes after 10 days (T10) from irradiation. LFC values at T3 (**C**) and T10 (**D**) are graphically plotted as a function of doses. Underlines numbers are the adj. *p*-value < 0.05 relative to D6 vs CTRL.

**Figure 7 cancers-16-01471-f007:**
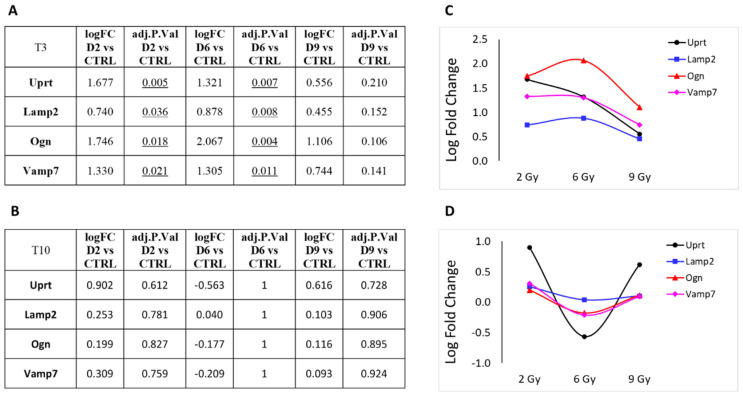
Protein-coding genes involved in cellular metabolism or vesicle trafficking pathways that showed significant over- or under-expression following exposure to 2, 6 or 9 Gy after 3 days (T3) from irradiation (**A**). (**B**) Same genes after 10 days (T10) from irradiation. LFC values at T3 (**C**) and T10 (**D**) are graphically plotted as a function of doses.

**Figure 8 cancers-16-01471-f008:**
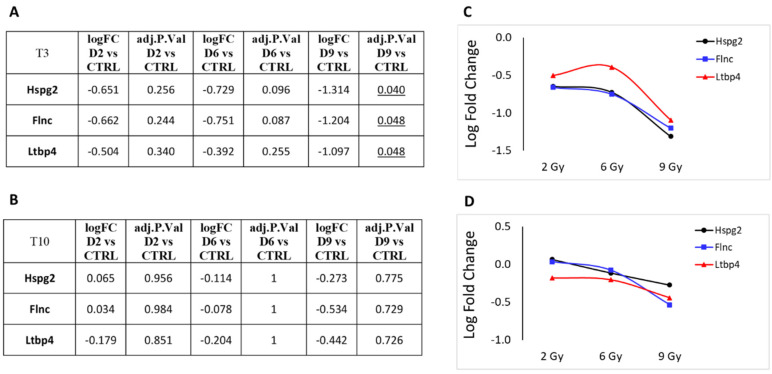
Protein-coding genes that play relevant roles specifically in muscle cells or in cardiac tissue that showed significant over- or under-expression following exposure to 2, 6 or 9 Gy after 3 days (T3) from irradiation (**A**). (**B**) Same genes after 10 days (T10) from irradiation. LFC values at T3 (**C**) and T10 (**D**) are graphically plotted as a function of doses.

**Figure 9 cancers-16-01471-f009:**
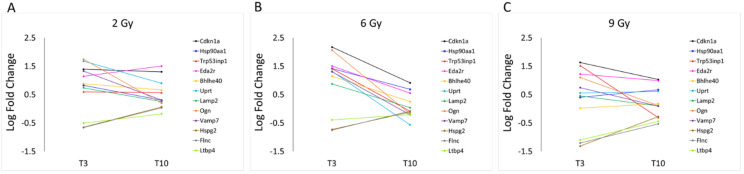
LFC values of protein-coding genes described in Appendix A, observed at 2 (**A**), 6 (**B**) and 9 Gy (**C**), are graphically plotted as a function of time: T3 and T10.

**Table 1 cancers-16-01471-t001:** Functional enrichment analysis of 2 Gy over-expressed samples: pathways affected by the over-expressed genes at 2 Gy dose. Legend: GO:MF = Gene Ontology, molecular function, GO:BP = Gene Ontology, biological process, REAC = Reactome database.

Source	Term_Name	Term_ID	Adjusted_*p*_Value	Intersections
GO:MF	RNA polymerase II transcription regulatory region sequence-specific DNA binding	GO:0000977	0.002695	*GM14295*, *ZFP970*, *GM14403*, *GM14322*, *ZFP971*, *GM14393*, *GM14399*, *GM14325*, *GM14326*, *ZFP931*, *ZFP965*, *GM14305*, *ZFP933*, *ZFP759*, *ZFP960*, *ZFP97*, *GM2026*, *ZFP938*, *ZFP935*, *ZFP729A*, *BHLHE40*
GO:MF	DNA-binding transcription factor activity	GO:0003700	0.000000	*GM14295*, *ZFP970*, *GM14403*, *GM14322*, *ZFP971*, *GM14393*, *GM14399*, *GM14325*, *GM14326*, *ZFP931*, *ZFP965*, *GM14305*, *ZFP933*, *ZFP759*, *ZFP960*, *ZFP97*, *GM2026*, *ZFP938*, *ZFP935*, *ZFP729A*, *BHLHE40*
GO:BP	regulation of transcription by RNA polymerase II	GO:0006357	0.001250	*GM14295*, *ZFP970*, *GM14403*, *GM14322*, *ZFP971 GM14393*, *GM14399*, *GM14325*, *GM14326*, *ZFP931*, *ZFP965*, *GM14305*, *ZFP933*, *MOSPD1*, *ZFP759*, *ZFP960*, *ZFP97*, *GM2026*, *ZFP938*, *ZFP935*, *ZFP729A*, *BHLHE40*
REAC	Gene expression (Transcription)	REAC:R-MMU-74160	0.000409	*GM14322*, *ZFP971*, *GM14325*, *ZFP931*, *CDKN1A*, *GM2026*, *ZFP938*, *GTF3C6*, *ZFP729A*

**Table 2 cancers-16-01471-t002:** (**a**) Functional enrichment analysis of 6 Gy over-expressed samples: pathways affected by the over-expressed genes at 6 Gy dose (D6). Legend: GO:MF = Gene Ontology, molecular function, GO:BP = Gene Ontology, biological process. (**b**) Functional enrichment analysis on 6 Gy under-expressed samples. Pathways affected by the under-expressed genes at 6 Gy dose (D6). Legend: GO:CC = Gene Ontology, cellular component, GO:BP = Gene Ontology, biological process.

(a)				
Source	Term_Name	Term_ID	Adjusted_*p*_Value	Intersections
GO:MF	RNA polymerase II transcription regulatory region sequence-specific DNA binding	GO:0000977	0.000569582	*GM14393*, *GM14399*, *GM14325*, *GM14326*, *ZFP931*, *ZFP965*, *GM14305*, *ZFP933*, *BHLHE40*, *DBP*, *PER1*, *FOS*, *ZFP759*, *ZFP729A*, *ZFP729B*, *ZFP960*, *ZFP97*, *MAX*, *XBP1*, *ZFP955B*, *ZFP760*, *ZFP953*, *ZFP935*, *ZFP72*, *ZFP712*, *ZFP273*, *ZFP938*, *ZFP433*, *FP930*, *ZFP975*, *ZFP84*
GO:MF	Transcription cis-regulatory region binding	GO:0000976	0.00256767	*GM14399*, *GM14326*, *GM14325*, *ZFP931*, *ZFP965*, *GM14305*, *ZFP933*, *BHLHE40*, *DBP*, *PER1*, *FOS*, *ZFP759*, *ZFP729A*, *ZFP729B*, *ZFP960*, *ZFP97*, *MAX*, *XBP1*, *ZFP955B*, *ZFP760*, *ZFP953*, *ZFP935*, *ZFP72*, *ZFP712*, *ZFP273*, *ZFP938*, *ZFP433*, *ZFP930*, *ZFP975*, *ZFP84*, *M14393*
GO:BP	Cellular amide metabolic process	GO:0043603	0.00049512	*RPS15*, *PER1*, *RPL38*, *ACOT2*, *ACOT10*, *RPL17*, *IMPACT*, *RPL15*, *PDHB*, *ABHD4*, *RPS29*, *DLD*, *HMGN5*, *EIF2S3X*, *RBM3*, *GSTA4*, *ABCE1*, *PSENEN*, *EIF3K*, *NGRN*, *MCEE*, *EIF4E3*, *GSTK1*, *RPL29*, *MRPS17*, *SCP2*, *GSTM4*, *GBA*, *EIF2A*
GO:BP	Establishment of protein localization	GO:0045184	0.019966747	*HSP90AA1*, *CDKN1A*, *PPP1R3C*, *ATAD1*, *ANXA1*, *Y IPF5*, *CRIPT*, *APOD*, *SNAP29*, *STK3*, *ENY 2*, *BTF3*, *BCAP29*, *VPS25*, *IFT20*, *XBP1*, *MDM2*, *PEX3*, *PTTG1IP*, *RAB9*, *SNX12*, *LAMP2*, *EMD*, *TIMM8B*, *VPS35*, *FOLR2*, *RAB6A*, *GOLT1B*, *EXOC4*, *CHMP5*, *SSR3*, *UFM1*, *SEC62*
GO:BP	Oxidative phosphorylation	GO:0006119	0.021330016	*COX7A2L*, *NDUFB9*, *COX7C*, *UQCRB*, *DLD*, *SDHD*, *RHOA*, *STOML2*
KEGG	Oxidative phosphorylation	KEGG:00190	0.039889377	*COX7A2L*, *ATP6V0E*, *NDUFB9*, *COX7C*, *UQCRB*, *SDHD*, *ATP6V0E2*, *NDUFC1*
KEGG	Drug metabolism—cytochrome P450	KEGG:00982	0.043325466	*MAOB*, *GSTA4*, *GSTK1*, *UGT2B38*, *UGT2B5*, *GSTM4*
**(b)**				
**Source**	**Term_Name**	**Term_ID**	**Adjusted_*p*_Value**	**Intersections**
GO:BP	Cellular response to radiation	GO:0071478	0.008787	*SWI5*, *MTCH2*, *HSPA5*, *IFI207*, *COPS9*
GO:CC	Myelin sheath	GO:0043209	0.037942	*TUBB4B*, *ATP5C1*, *CD59A*, *HSPA5*
KEGG	Chemical carcinogenesis—reactive oxygen species	KEGG:05208	0.033798	*NFE2L2*, *NDUFA8*, *ATP5C1*, *EPHX1*

**Table 3 cancers-16-01471-t003:** Functional enrichment analysis of 9 Gy over-expressed samples: pathways affected by the over-expressed genes at 9 Gy dose (D9). Legend: GO:MF = Gene Ontology, molecular function.

Source	Term_Name	Term_ID	Adjusted_*p*_Value	Intersections
GO:MF	DNA-binding transcription factor activity, RNA polymerase II-specific	GO:0000981	0.0008	*ZFP971*, *GM14393*, *GM14325*, *ZFP931*, *ZFP965*, *GM14305*, *ZFP960*
GO:MF	DNA-binding transcription factor activity	GO:0003700	0.0011	*ZFP971*, *GM14393*, *GM14325*, *ZFP931*, *ZFP965*, *GM14305*, *ZFP960*
GO:MF	Transcription regulator activity	GO:0140110	0.0071	*ZFP971*, *GM14393*, *GM14325*, *ZFP931*, *ZFP965*, *GM14305*, *ZFP960*
KEGG	Proteoglycans in cancer	KEGG:05205	0.0312	*HSPG2*, *FLNC*, *CDKN1A*

## Data Availability

The datasets used and/or analyzed during the current study are available from the corresponding authors on reasonable request.

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
