# Peer review of "Impact on the Transcriptome of Proton Beam Irradiation Targeted at Healthy Cardiac Tissue of Mice"

_cancers, 2024, doi:10.3390/cancers16081471_

Round 1

Reviewer 1 Report

Comments and Suggestions for Authors

Cancers, MDPI, 03_2024

Dear Editor and authors,

the manuscript “Impact on the transcriptome of proton beam irradiation targeted to healthy cardiac tissue of mice” describes transcriptome changes in mice heart tissue after proton irradiation. The analysis includes 2, 6, and 9 Gy at days 3 and 10 after exposure and a pathway analysis of deregulated genes. Concerning proton irradiation as an emerging treatment modality, this study provides mechanistic data on an important topic for optimal treatment and protection of radiotherapy patients. However, the manuscript is purely descriptive, without any attempt to confirm findings. The discussion includes a lot of repetition of results but lacks interpretation and comparisons to the existing literature. Therefore I suggest major revisions.

-          in terms of radiation-induced effects on the cardiovascular system it is not correct to assign 10 days as a late effect

-          results for T10 are surprising and potential experimental origins should be discussed

-          how is the dose distribution confirmed?

-          bell-shaped dose-response is an interesting finding, however, the selected microarray data should be confirmed by an alternative method (e.g. CDkn1a, Hsp90 …… )

-          according to Fig 1 and l.182 there is only one significantly regulated gene after T10, thus the logFC in Fig 5BD, 6BD and 7BD after T10 is not relevant

-          Fig 5BD check data for Cdkn1a

-          Fig 5, 6, 7 legends, please correct “ following exposure to both 2, 6 and 9 Gy….”. The pathway should replace group 1, 2,3

-          l. 345 “…, almost all genes showed a significant variation.” According to Fig.1 only one gene is significantly regulated, please clarify

-          l.350 – 353, what does this statement mean?

-          the starting material should be specified, whole hearts?

-          l.428 how can the regulation in heart tissue display radiotherapy efficacy? maybe radiotherapy side effects?

-          correct y- axis labelling in Fig 2, 3, 4, what is the rationale for the different sequence of dose and number of genes in the headline?

-          the selection of genes for Fig 2, 3,4 is not well described

Comments on the Quality of English Language

english is fine with few minor editing

Reviewer 2 Report

Comments and Suggestions for Authors

The authors provide a mostly descriptive accounting of their experience with developing an animal model for proton cardiac irradiation. They provide summary results of the gene expression changes relative to controls at multiple timepoints and dose levels. The analysis is comprehensive, but the story hasnt been developed and my guess is that there is more that can be done with the data by revising the way the data is presented and digging into some of the more notable changes in genes/pathways that are connected to radiation related cardiac dysfunction. I suggest refining the figures/tables such that they are presented in a way that improves the reader's discernment of which biologic processes are most relevant at the doses investigated in the animal model and connect these to key genes. As it stands, the data is presented in a way that is mostly descriptive. The introduction, methods/materials, results are well written, but the interpretation of the experimental data in the tables/figures/discussion is not fully realized and is thus not connected to current disease models of RT related cardiac dysfunction. 

In accordance with the journal’s standards, and their adoption of FAIR principles, the data and code should be made available as a supplement or linked to in a publicly accessible database.

Simple Summary:

-Suggest Rephrasing: “Different types of ionizing radiation have different effects on biological targets. While most available data on radiotherapy comes from studies using photon sources, other types of radiation, like protons, have distinct physical properties that can lead to different biological outcomes. Extensive research supports this, but more work is needed, especially on the effects on healthy tissues surrounding tumors treated with different radiation types. This study investigates the effects of proton therapy on healthy cardiac tissue by analyzing gene expression changes at different radiation doses and time points.”

Abstract:

-No changes suggested

Introduction:

-Replace "peculiar responses" with "unique effects" for clarity.

Methods and materials:

-Replace “hearths” with hearts on line 124 page 3

Results

-Well written.

Discussion:

-Line 341-353 - consider cutting or reducing the size of this paragraph as it summarizes the results in a descriptive fashion and does not advance the author’s story.

-Line360-398 - this paragraph should be expanded and possibly split up as this is really where the bulk of the knowledge gained from these experiments becomes relevant. 

-Consider adding another paragraph talking about the advantages and disadvantages of using alternative techniques beyond bulk gene expression i.e. scRNAseq and possibly looking at the epigenetic changes at longer time points i.e. T10 using the infinium 450K platform.

Figures / Tables:

An overall experimental schema representing the # animals, exposures, timepoints and analyses are recommended. Something akin to : https://bit.ly/48GkT9E or https://bit.ly/3v4Vusu would be beneficial.

Figure 2: please correct the spelling in the y-axis label “Cahnge”

-It might be helpful to sort the DEGs in Fig2 by max or min fold change or maybe by gene ontology group, at present the random arrangement doesnt help the author tell their story and it is purely descriptive. I.e. no hypothesis is being confirmed or refuted with this figure. Consider vertically orienting the figure such that the x axis is fold change for all the up regulated genes. The 2 genes which are downregulated could either be discussed in the text or shown in a separate smaller subfigure.

Figure 3: please correct the spelling in the y-axis label “Cahnge”

-Consider vertically orienting the figure such that the x axis is fold change for all the up regulated genes.

-Once reoriented, the 2 bargraphs could be joined rather than the current depiction which is arbitrarily separated into to similar plots.

Figure 4: please correct the spelling in the y-axis label “Cahnge”

-Consider vertically orienting the figure such that the x axis is fold change for all the up regulated genes.

-The 3 genes which are downregulated could either be discussed in the text or shown in a separate smaller subfigure.

Table 1,2,3. 

-Consider combining these tables into one and add a column to indicate which genes and GEO were perturbed at each dose level and in what direction. As it stands, these 3 tables are entirely descriptive and dont tell a story related to the authors hypothesis.

-I would sort the LFC for the genes in Fig 2, 3, and 4 by these gene ontology groups

Table 4.

-This is useful, but I am not sure it should be a manuscript table, i.e. consider putting it in the supplement, and referring to it in the text when necessary.

Figure 5, 6, 7

-These are probably the most informative figures in the paper, but only one timepoint is examined and instead dose is represented on the x axis. I.e. the log fold change is modelled by dose, when I think it would be more biologically informative to model the LFC for protein coding genes over time and stratify the modelled genes by dose to see if the degree of injury or biological process are substantially different. I.e. it might be just as relevant to do a version of fig 5 and 6 for RT - CTRL fold change irrespective of dose, and plot LFC on the y axis and time on the x-axis. If trends emerge, then it may be that the differences in dose-levels may help providers infer the threshold dose for that type of injury. 

-We need to see more about which genes had persistent LFC changes relative to baseline that lasted at T10. Are they related to pathways that are likely involved in chronic pathology/heart dysfunction?

Comments on the Quality of English Language

There are a few areas where the quality of the language/writing could be improved. I would suggest some modern techniques for rephrasing/rewriting like gemini, chatgpt, copilot or something similar.

Round 2

Reviewer 1 Report

Comments and Suggestions for Authors

Dear editor and authors,

thank you very much for the improvements. In my view the manuscript is now ready for publication.

minor comments:

l.28 it should be "studies"

l.33 the wording of the first sentence should be reconsidered (e.g. compared to is better than in respect)